# Animal Coronaviruses and SARS-COV-2 in Animals, What Do We Actually Know?

**DOI:** 10.3390/life11020123

**Published:** 2021-02-05

**Authors:** Paolo Bonilauri, Gianluca Rugna

**Affiliations:** Experimental Institute for Zooprophylaxis in Lombardy and Emilia Romagna—IZSLER, 25124 Brescia, Italy

**Keywords:** SARS-CoV-2, animals, veterinary

## Abstract

Coronaviruses (CoVs) are a well-known group of viruses in veterinary medicine. We currently know four genera of Coronavirus, alfa, beta, gamma, and delta. Wild, farmed, and pet animals are infected with CoVs belonging to all four genera. Seven human respiratory coronaviruses have still been identified, four of which cause upper-respiratory-tract diseases, specifically, the common cold, and the last three that have emerged cause severe acute respiratory syndromes, SARS-CoV-1, MERS-CoV, and SARS-CoV-2. In this review we briefly describe animal coronaviruses and what we actually know about SARS-CoV-2 infection in farm and domestic animals.

## 1. Introduction

The *Coronaviridae*, with a single-stranded, positive-sense RNA genome is a well-known and studied family of viruses in veterinary medicine. Virtually every pet, breed, or companion animal and every wild animal can be said to have dealt with at least one virus from this family in its life. Coronaviruses (CoVs) are currently divided in four genera: *Alpha-*, *Beta-*, *Gamma-*, and *Deltacoronavirus*, and all genera are of interest as etiologic agents of enteric, respiratory, or systemic diseases in animals. The most common wild animal host of coronaviruses is bat, and it is commonly accepted that a family of virus associated to severe respiratory syndrome (SARS), named SARSr-CoV, is mainly found in bats. It had been expected that a future disease outbreak would come from this family of viruses [1,2].

SARS-CoV-2 is only the latest example of an emerging zoonotic infectious virus that has become pandemic after the spillover. Also in that pandemic, the bat was the most likely candidate to be the origin of this virus, while the intermediate host was only hypothesized [3]. The first case of SARS-CoV-2 infection was reported in Wuhan, China, on 31 December 2019, with symptoms of atypical severe pneumonia [4,5]. This case was further confirmed to be caused by the novel coronavirus that, according to the WHO, as of 6:23pm CET, 14 December 2020, has caused 71,051,805 confirmed cases of COVID-19, including 1,608,648 deaths worldwide [6].

The last human coronavirus is the seventh coronavirus that achieved the capacity to infect humans: SARS-CoV, MERS-CoV, and SARS-CoV-2 can cause severe disease, whereas HCoV-HKU1, -NL63, -OC43, and -229E are associated with mild symptoms. The four coronaviruses of common cold (HCoV-229E, -NL63, -OC43, and -HKU1) had an evolutionary history and host associations that helped to convince scientists and the general population that the spillover of coronavirus had happened many times in the past. Research on this family of viruses can provide important insights into the natural history of past human pandemics [7]. The last one before the actual pandemic spillover of coronavirus was recorded historically at the end of the 19^th^ century [6], when the HCoV-OC43 was speciated from the bovine coronavirus that has been suggested to be a possible ancestor in recent phylogenetic analysis [7].

Now, the world is fighting to mitigate the consequences on human health and to survive in the socioeconomic crisis that the not-sanitary measures applied around the world to flatten the epidemic curve of COVID-19 has brought them. In addition, SARS-CoV-2 is demonstrating the ability to infect and sometimes cause respiratory distress in many mammalian species. Transmission from humans to dogs, cats, lions, tigers, and minks has occurred, and in the last case, mink-to-mink intraspecific transmission was observed and notified to OIE in many countries [8]. The involvement of different mammalian species, that are domestic, farmed, and wild animals in SARS-CoV-2 circulation has potentially dramatic implication and indicates the need for One Health surveillance, intervention, and management strategies to mitigate the effects of this potential panzootic virus [9].

For this reason, beginning with this knowledge, we will try to summarize in this review organized by large chapters what we know about animal coronaviruses and how the current pandemic virus can infect and sometimes create disease among animals, while trying to draw useful insights also concerning the trend of the pandemic in humans.

## 2. Animal Coronaviruses

Coronaviruses (CoVs) are enveloped viruses with a positive-sense, single-stranded nonsegmented RNA (ssRNA) genome (26–32 kb). According to the current taxonomy, CoVs are classified as one of the two genera in the subfamily Coronavirinae, family Coronaviridae of the order Nidovirales [10]. Coronaviruses (CoVs) are sorted in four genera on the basis of their phylogenetic relationships and genomic structures: *Alpha-*, *Beta-*, *Gamma-,* and *Delta-CoV*. Almost all alpha-CoVs and beta-Covs have mammalian hosts, while gamma-CoVs and delta-CoVs are commonly found in avian hosts, even though some of them can also infect mammals. Members of this large family are considered as causative agents of respiratory, enteric, hepatic, and neurological diseases in birds and mammals. The CoVs of interest in veterinary medicine are reported here and subdivided by host species of interest (see also Table 1).

### 2.1. Coronaviruses of Birds

Avian CoVs belong to the genus *Gammacoronavirus* that includes three major species: infectious bronchitis virus (IBV), pheasant coronavirus (PhCoV), and turkey coronavirus (TCoV). IBV or IBV-like gammacoronaviruses have been found in other avian species such as peafowl, partridge, blue-winged teal, pigeon, guineafowl, and various wild bird species [11].

The infectious bronchitis virus, the first coronavirus discovered, is to date the most important and best-studied gammacoronavirus and is therefore considered the genus’ prototype. This virus is of great economic importance to the poultry industry worldwide, affecting the performance of both meat-type and egg-laying birds. IBV is a highly contagious disease that affect the respiratory, reproductive, and renal systems, with a severity that differs depending on the involved viral strain [12].

### 2.2. Coronaviruses of Domestic Carnivores

Two coronaviruses are known in dogs: two alphacoronaviruses, namely CCoV-I and CCoV-II, and the betacoronavirus CRCoV. CCoVs are gastrointestinal viruses with fecal-oral transmission that are commonly observed in dogs all over the world but in most cases, are considered to cause a very mild gastrointestinal disease in pups or are completely asymptomatic [13]. However, a recently characterized strain (CB/05) resulted in a fatal disease as a consequence of systemic spread of the virus [14]. Moreover, intestinal villi infected by CCoV seem to enhance cells susceptibility to canine parvovirus (CPV) infection. This causes a synergistic action that ended in a much more serious disease than the one that both viruses can cause separately [15]. Unlike CCoVs, the betacoronavirus CRCoV, also known as canine coronavirus group II, causes mild respiratory signs in dogs and is considered the etiological agent of canine infectious respiratory disease (CIRD) together with other viral and bacterial agents [16]. This beta-CoV is generically related to one of the human coronaviruses of common cold, namely HCov-OC43, and to bovine coronavirus BCoV [17].

Feline coronaviruses FCoV-type I and FCoV-type II belong to genus *Alphacoronavirus*. Both genotypes cause a mild enteric disease that, in most infected cats, show no signs of disease. However, in a percentage of cases the enteric coronaviruses can, within the host, undergo a mutation and acquire the ability to infect monocytes/macrophages, causing a systemic disease [18]. In that form, named feline infectious peritonitis (FIP), the virus causes a serious disease related to an intense immune response, with a fatal outcome in most of the cases.

### 2.3. Coronaviruses of Swine

Six CoVs can cause infection in swine. These include four alphacoronaviruses, namely transmissible gastroenteritis virus of swine (TGEV), porcine respiratory coronavirus (PRCoV), porcine epidemic diarrhea virus (PEDV) and SADS-CoV, one betacoronavirus, namely porcine haemagglutinating encephalomyelitis virus (PHEV), and one deltacoronavirus, the porcine deltacoronavirus (PDCoV) [19]. TGEV, PEDV, SADS-CoV, and PDCoV are responsible for acute gastroenteritis in swine. PRCoV causes a mild respiratory disease, and PHEV is the causative agent of neurological and/or digestive disease in pigs.

### 2.4. Coronaviruses of Bovine

The most common coronavirus of bovine is BCoV. This virus is able to cause a variety of clinical forms, including a severe enteric disease in neonate calves, winter disease (a severe enteric form) in adult dairy cattle, and a respiratory disease in cattle of all age groups [20].

Interestingly, HCoV-OC43 likely evolved from ancestral BCoV strains that crossed the interspecies barrier and established an infection in human beings around 1890, following a 290-long nucleotide deletion downstream of the spike gene [21].

### 2.5. Coronaviruses of Horse

The only CoV that has been so far discovered in horses is the equine coronavirus ECoV, which belongs to the genus *Betacoronavirus*. ECoV is a newly recognized enteric virus of adult horses that has been associated with fever, lethargy, and anorexia, as well as colic and diarrhea. Outbreaks have been reported in Japan, Europe and the USA since 2010 [22].

## 3. Molecular Basis of SARS-CoV-2 Infectivity

SARS-CoV-2 uses the envelope-embedded surface-located spike (S) glycoprotein to interact with cells expressing the angiotensin-converting enzyme 2 (ACE2) receptor [4]. The S glycoprotein comprises two functionally distinct subunits, namely S1 and S2, which are involved in receptor recognition and membrane fusion, respectively [23]. The interaction between ACE2 and SARS-CoV-2 S protein involves a C-terminal domain of the S1 subunit, also called receptor-binding domain (RBD) [24], that is a key determinant of viral infectivity and host specificity.

The susceptibility of different animal species to SARS-CoV-2 is of concern due to the potential for interspecies transmission, and its understanding is crucial for controlling the virus spread. Indeed, coronaviruses exhibit a strong cross-species transmission potential, due to their genetic adaptive variation (Figure 1), especially when it involves the RBD [25]. For example, two amino acid residues in RBD were identified to be essential for SARS-CoV transmission from civets to humans [26]. That said, it is important to understand whether animals may become infected and potential transmitters to humans.

The host range of SARS-CoV-2 may be extremely broad due to the expression of ACE2 in a large spectrum of vertebrate animals. However, variation in critical ACE2 residues involved in RBD/receptor interaction may influence the susceptibility of different species to this CoV. Thus, a comparative analysis of ACE2 protein sequences can be used to predict their affinity for SARS-CoV-2 S glycoprotein binding, and as a consequence, the species that may serve as a reservoir host for this virus.

Li and colleagues [39] first performed amino acid sequence alignment of ACE2 from different species, including human, five nonhuman primates, eight domestic animals (cat, dog, bovine, sheep, goat, swine, horse, and chicken), three wild animals (ferret, civet, and Chinese horseshoe bat), and two rodents (mouse and rat). Authors found that human and nonhuman primates share identical sequences in some regional residues. The high sequence similarity observed in most companion, domestic, and wild animal imply that ACE2 from these animals may recognize SARS-CoV-2; thus, these animals may be susceptible to the infection. On the other hand, rodents and chickens are not likely susceptible hosts.

More recently, Damas and colleagues [40] employed a combination of comparative genomic approaches and protein structural analysis to study the conservation of ACE2 from 410 vertebrate species and its potential to be used as a receptor by SARS-CoV-2. This study confirmed the high susceptibility of primates to SARS-CoV-2. On the other hand, this study predicted a broader group of species that may serve as a reservoir or intermediate host for this virus, with mammals having a medium-to-high score for the propensity of their ACE2 to bind with the SARS-CoV-2 S protein.

Based on findings from molecular studies, the ACE2 proteins of nonhuman primates and most of the companion, domestic, and wild animals closely resemble the human ACE2 receptor. However, data may show discrepancies between the predicted susceptibilities to infection and those experimentally observed. Indeed, these studies are based solely on in silico analyses and are based on a small number of amino acid residues, i.e., twenty-five amino acids corresponding to known SARS-CoV-2 S-binding residues [40]. Cross-species transmission does not rely solely on the presence of the receptor but also on the levels of ACE2 expression in the respiratory mucosa and on the presence of other cellular factors required for viral replication. Therefore, these studies need validation by experimental infection in animal models or examples of real-world infections.

## 4. Case Studies in the Context of SARS-CoV-2

### 4.1. Experimental Infections in Animals

Dogs and cats have been extensively explored to study the SARS-CoV-2 transmission, due to their close relationship with human beings. The first published experimental infection involving cat [41] showed that cats could become infected by SARS-CoV-2 and potentially transmit the virus to other cats via the airborne route. Both RT-qPCR and immunohistochemistry assays revealed abundant RNA or antigen in respiratory and gut epithelium. These findings were confirmed in other studies [42,43] showing that infected cats shed the virus for no more than 6 days following exposure, suggesting that cats will develop and clear infection rapidly. In contrast, five-months-old beagles were intranasally inoculated with SARS-CoV-2, and it was found that only the directly inoculated animals demonstrated viral RNA only in rectal swabs 2 day’s post infection [41]. None of the analyzed tissues showed detectable virus particles or viral genomes, which indicate that dogs have a low susceptibility.

The experimental infection of ferrets has been described in different studies, since their lungs share many similarities with those of humans. The virus was shown to replicate efficiently in the upper-respiratory tract of these animals [41,44]. Moreover, two studies provided the experimental evidence that SARS-CoV-2 can be transmitted efficiently via direct-contact and via the air between ferrets, resulting in a productive infection and the detection of infectious virus in indirect recipients [45,46].

A recent preprint not-peer-reviewed publication on experimental infection of rabbits (*Oryctolagus cuniculus*) demonstrated no obvious clinical signs. However, viral RNA was present in the nasal, throat, and rectal swabs, suggesting the susceptibility of rabbits to SARS-CoV-2 [47].

Under experimental conditions, cattle show low susceptibility to SARS-CoV-2 infection [48] corresponding with a predicted medium susceptibility of cattle species on the basis of a computational modeling of their ACE2 receptor [40]. Experimental studies using other animals such as chicken and duck demonstrated that these avian species are not susceptible to SARS-CoV-2 infection [41,44].

Studies on pigs are inconsistent. While previous studies [41,49] found that pigs seem to be resistant to SARS-CoV-2 infection and are unlikely to be a SARS-CoV-2-carrier animal species, a recent experimental infection contradicted these findings, suggesting that infectious dose and host-related factors could influence the experimental outcome [50].

Finally, to date, there are knowledge gaps on the role of other farmed animals, such as sheep, goats, and horses.

### 4.2. SARS-CoV-2 Serological Surveillance in Animals

There are surprisingly only two serological surveys conducted in domestic animals published or in preprint at the time of this review. The first serological survey involved cats in Wuhan (China), considered to be the city where the epidemic began [51], where a cohort of 141 serum samples were considered. Thirty-nine sera were collected between March and May 2019 before the COVID-19 outbreak, and 102 between January and March 2020 after the outbreak. Sera sampled from 46 abandoned cats were from three animal shelters, 41 cats were from five pet hospitals, and 15 cats were from COVID-19 patient families. All the sera were analyzed by indirect enzyme-linked immunosorbent assay (ELISA) and virus neutralization assay (VNT). Indirect ELISA found a positive reaction for the receptor binding domain (RBD) of SARS-CoV-2 in 15 (14.7%) sera of cats collected after the outbreak. FIPV hyperimmune sera was used as negative controls and did not show crossreactivity with SARS-CoV-2. Eleven out of 15 ELISA positive sera (73%) tested positive in VNT. In particular, two cats were from the same owner who was a COVID-19 patient and have shown high specific neutralization titer. To track the dynamic characteristic of serum antibody against SARS-CoV-2 in these felids, the same two cats were sampled every 10 days over 130 days, and the authors found that RBD antibodies of these two cats reached the peak at the second sampling (recent infection) and titer declined progressively until 110 days, when the level of antibodies was under the detection level of tests (ELISA and VNT). The authors discussed the results as a demonstration that that SARS-CoV-2 had infected cats in Wuhan during the outbreak and provided useful information on the dynamics of serum antibodies in cats. It is worthy to note that the duration of serum antibodies in cats is not particularly different in respect to what is observed in humans with mild symptoms of the disease [52].

The second serological survey was conducted in Italy [53] and was a large-scale study mostly focused in Lombardy, the Italian region affected by the highest circulation of SARS-CoV-2 during the first wave. Nine-hundred-and-nineteen companion animals (cats and dogs) living in northern Italy were sampled at the time of frequent human infection. All animals were sampled by their private veterinary surgeon during a healthcare visit. A total of 603 dogs and 316 cats were sampled. Moreover, oropharyngeal (303 dogs, 173 cats), nasal (183 dogs, 78 cats), and/or rectal (66 dogs, 30 cats) swabs were collected from a total of 494 animals. SARS-CoV-2-neutralizing antibodies were detected in 15 dogs (3.3%, 15/451) and 11 cats (5.8%, 11/191), with higher titers observed in cat than dog serum samples. Among animals, dogs living in contact with positive COVID-19 people have significantly higher risk to be positive (odd ratio 8.5; 95% CI: 1.7 to 43.5 *p* < 0.05 recalculated for data presented in [53]); this association was not confirmed in cats. Male dogs have a four times higher risk to be positive (odd ratio 4.4; 95% CI: 0.9 to 21.2 recalculated from data presented in [53]), but in that case the statistical significance of the ratio was not achieved (*p* > 0.05). Interestingly, a positive direct correlation was observed in Lombardy provinces between the proportion of dogs that tested positive and the recorded burden of human disease. A similar association was observed for cats. Neither regression analyses reached significant levels (*p* > 0.10). All animals tested negative by a molecular swab test regardless of the type of swab (oropharyngeal, nasal, or rectal) or living in a positive household with confirmed COVID-19 humans. Authors conclude that although higher antibody titers were detected in cats, which was surprisingly not previously reported, dogs need more investigation regarding their SARS-CoV-2 susceptibility because of the higher risk observed when dogs live in COVID-19-positive houses, and the association to male sex seems to be similar to what is observed in humans [53]. This extensive survey suggests that infection in companion animals has to be considered not unusual.

### 4.3. SARS-CoV-2 Notified Cases in Animals Structured by Countries

We then proceed with animal cases notified to the OIE (World Organization for Animal Health) [8] to date, subdivided by country, following the temporal order of the notification in the year 2020. A summary table of notified cases and a world map of cases of infection in animals can be found on the OIE website at the following link https://www.oie.int/en/scientific-expertise/specific-information-and-recommendations/questions-and-answers-on-2019novel-coronavirus/events-in-animals/ (accessed on 15 January 2020 at 3:10 p.m.). A specific subchapter will be reserved for minks.

In Hong Kong, the first two cases notified were two infected dogs in Hong Kong on February 26. The dogs belonged to a person hospitalized, and for this reason, the animals were placed in a quarantine clinic. They never showed symptoms, but they tested positive from both nasal and oral swabs on February 28 and in the following three tests at a distance of 2 or 3 days from each other, until a complete viral cleaning on March 13. These two dogs from the peninsula located in southeast China were joined by five cats, all identified in the context of veterinary epidemiological surveillance linked to human cases and the last update notified was on November 27.

In Belgium, on March 28, the Belgian National Veterinary Service notified a case of a cat that tested positive for SARS-CoV-2 in gastric contents and feces samples. The analysis was conducted in the Faculty of Veterinary Medicine of the University of Liege. The cat exhibited gastrointestinal (diarrhea and vomiting) and respiratory (cough and dyspnea) clinical symptoms. The cat belonged to a positive person who became infected during a trip and was therefore quarantined [54]. Subsequent analyses did not clarify whether it was an active infection, but the laboratory data (Ct of the PCR and sequencing of RNA) and the clinical signs were considered compatible with SARS-CoV-2 infection. The last update notified was on March 28.

In the USA, the first reported case of transmission in animals involved big cats from the Bronx Zoo, five tigers and three lions. The first animal to show any symptoms was a tiger on March 27, but on April 3, three other tigers and all three lions showed respiratory symptoms (dyspnoea and dry cough), and one animal lacked appetite. Two different SARS-CoV-2 genotypes were identified in lions and tigers, respectively, [55] that testify that there have been at least two different humans as sources for big-cats passages. Furthermore, the virus was detected in the felines both in the respiratory secretions and in the feces, indicating active replication and the possibility of spreading the virus by these animals. The episode of the Bronx Zoo ended without serious complications in the felines, which were all transferred to other accommodation, and in any case, no other cases were subsequently observed besides the four tigers and three lions involved. In addition, in the USA, cases were observed in cats and one dog owned by positive or COVID-19 sick people. A total of 14 cats from 11 states and 11 dogs from 8 states were sampled because animals came from a household with known COVID-19-affected inhabitants. In many cases, respiratory and gastroenteric signs was reported for cats and in some cases for dogs. In almost all cases, the animals were admitted to the veterinary service for different reasons and tested because they were owned by people who tested positive; indeed, active veterinary surveillance is not implemented in the United States. Moreover, another two Malayan tigers at a zoo in Tennessee exhibited clinical signs, including mild coughing, lethargy, and inappetence, and were confirmed positive for SARS-CoV-2 and a third tiger, who was also showing clinical signs was sampled and suspected. Even in the United States, there have been episodes in the minks that we will return to. The last update notified was on November 27.

France in May reported two suspected cases involving cats from owners previously suspected of being infected with COVID-19. For each cat, rectal and nasopharyngeal swabs were taken. One cat tested positive by qRT-PCR on a rectal swab but not from nasopharyngeal swabs. This cat showed mild respiratory and digestive signs. France also reported an outbreak in mink on November 25.

Spain also reported two cases in cats: the first symptomatic with pneumonia and breathing difficulties in April and the second asymptomatic, identified following veterinary surveillance of the pets of virus-positive people in May. Spain reported an outbreak in mink on July 16.

In Germany, there is only one report on a cat owned by a subject who died in April, cohabiting with two other cats, and which showed a weak positivity to the virus, remaining asymptomatic.

Russia reported a 5-year-old cat tested positive to the presence of SARS-CoV-2 in throat and nasal swabs, with more information closed as it was a case of unknown origin, resolved on the first of June.

In the United Kingdom, only one notification was reported for a domestic cat from a COVID-19-positive household. The cat was showing respiratory signs suggestive of feline herpes virus (FHV) infection but also tested positive to SARS-CoV-2. Confirmation of the case was obtained in July, when the cat tested positive to the virus neutralization test, and the case was considered resolved on July 28.

In Japan, four cases have been observed in dogs owned by people hospitalized or isolated in quarantine. These dogs were entrusted to a private pension service from late July to early August. Nasal and throat swabs were taken from the dogs and tested positive to SARS-CoV-2. None of these four dogs have exhibited any clinical signs. Also two cats that were entrusted to the private company for the same reason tested positive in nasal and throat swabs. The dogs and cats were returned to the owners after a negative result, and the last report was dated on November 6.

South Africa notified a case in a zoo puma in July. The animal tested positive for SARS-COV-2 after contact with an infected handler. All other animals in contact tested negative, and the case was solved on September 17.

Italy notified of a cat that tested positive in December in nasopharyngeal swabs but not in the rectal swab. The cat developed detectable antibodies in ELISA and immunochemiluminometric assay [IMCA]. On December 9, no data on cat anamnesis were reported.

Chile reported three cases in cats from May. The first animal was positive by RT-PCR in nasal secretions and feces. Four and five days after this finding, another two cats were found positive. Viral excretion in these three cats was observed for 4, 7, and 16 days, respectively. After that, the virus became undetectable and the report ended on October 22nd.

Canada notified the OIE on October 28 of the first detection of SARS-CoV-2 in an asymptomatic dog detected in the aim of a research study targeting companion animals from households with confirmed cases of COVID-19. The onset of the first human case in the household was approximately ten days before the dog positivity detection. Another dog was present in the household but tested uninfected.

In Brazil, a female domestic cat that cohabited with a dog in a household with COVID-19-confirmed human cases tested positive in oral and nasal swabs. The dog’s samples were negative, and the report was closed on October 29.

Argentina, starting from the first days of September, notified about two cats and four dogs. These samples were taken as part of research projects on pets living with people affected by COVID 19. Oropharyngeal and rectal swabs were taken, and sera were collected. In total, the country reported four outbreaks on a total of four suspected cats and 12 suspected dogs. Two cats showed symptoms of weakening and anorexia for 12 to 24 hours and sneezing and nasal secretions. One of the dogs showed conjunctivitis, cough, dyspnea, and weakening; the rest of the animals were asymptomatic. The final report was realized on November 16.

Switzerland recently announced the confirmation of SARS-CoV-2 in a domestic cat in the Canton of Zurich, detected in the frame of a research project on pet animals of owners who were infected with SARS-CoV-2. The cat that lived with another cat showed signs of an upper-respiratory infection including sneezing, inappetence, and apathy, and tested positive in virological and serological tests, while the cohabiting cat remained asymptomatic and negative. The first notification was released on December 3.

To summarize these real-world cases of infected animals around the world, we can observe that in all the cases where epidemiological data are available, domestic animals have a link to a human being with COVID-19. Everywhere outside the USA, animals were sampled because they were owned by COVID-19-positive people in the aim of veterinary active surveillance plans or research projects. In the USA, only passive surveillance was applied, and animals were sampled at veterinary clinics and were conferred because of the suffering of different signs. In any case, among domestic pets, cats were infected more frequently than dogs (35 cases were reported in cats and 22 in dogs), as predicted in molecular-receptor-affinity studies and experimental infections. Cats, in a remarkable number of cases (7 out of 21 cases reported from outside the USA and 11 out of 14 in the USA) were reported as symptomatic. Symptoms in cats are mild in most cases, and include signs of an upper-respiratory infection with sneezing as the main symptom reported, along with general signs such as loss of appetite, apathy, weakness, and sometimes an ocular discharge was observed. In two cases only, outside the USA, cats had suffered severe respiratory symptoms with bilateral pneumonia and breathing difficulties, which led to the euthanasia of the suffering animals. Dogs confirmed their lower susceptibility to the virus with 22 notifications globally and only one symptomatic case reported outside the USA.

### 4.4. Mink

SARS-CoV-2 has high affinity for Mustelidae’s ACE-2 receptor [56] and in the second part of the year 2020, the virus has demonstrated active and extensive circulation ability among American mink (*Neovison vison*) in nine European countries and in the USA. Herein, we report in detail the outbreaks in mink farms notified to OIE in 2020.

USA: Confirmed cases of SARS-CoV-2 in domestic mink have been identified so far on 17 commercial premises, most of which are located in Utah, one farm was in Wisconsin, and another one in Oregon, as reported in the 26^th^ follow-up report on November 26. Increased mortality rate, inappetence, and sudden death are the signs frequently observed before SARS-CoV-2 detection in farms. In one case, the probable transmission of mink-to-man was observed in a person who was in close contact with the animals on the farm. The American cases are mostly single events that involved a few farms and almost 90,000 animals in all, with the symptomatology returning in a short period. Up until now, in the USA, no killing of animals has been practiced on farms.

The Netherlands: The Netherlands was the first country to report SARS-CoV-2 in three farms in April 2020. The last available follow-up report was dated October 6 and counted 62 farms with infected animals. Clinical signs of respiratory disease, lack of appetite, and increased mortality were reported in 25 farms, whereas in 37 farms, the virus was detected as part of specific veterinary surveillance plans or through the early warning system (obligation to send in dead animals weekly). That farms are located in the neighborhood of symptomatic cases. Among the animals of these farms, many initially showed only weak positivity in PCR, followed by clinical signs after a few days. This testifies to the excellent rapid-alert system and tracking of positive farms in this country. In other cases, the animals that were slaughtered following detection of the virus were found to be seronegative, indicating an infection so recent that it did not significantly circulate among the animals. However, in some farms up to 70% of the animals tested positive for antibodies, indicating a not-recent infection at the time of the detected positivity. The Dutch health authorities believe that the cases in mink farms occurred despite all the operators having correctly implemented measures to prevent the spread of the virus to animals for several months: the hypothesis put forward is that the virus arrived in the farm from other wild animals (bats, mustelids, and wild birds) [57].

France: Positivity among animals was notified on November 16. In a farm of 4,100 animals, blood and swabs were taken from 180 animals. 174 animals (96.6%) were positive on the serological test and 33 out of 110 on the virological test (real-time RT-PCR). All the animals were slaughtered and destroyed. Investigations are underway on neighboring farms and no follow-up reports were released after the first notification.

Spain: Since the first positive communication in the Netherlands, the Spanish government increased clinical surveillance in mink farms, and in the second fortnight of May, a suspicion was reported in a mink farm in the municipality of Puebla de Valverde, province of Teruel. In this case, none of the workers at the same company tested positive for the virus, and none of the animals showed symptoms consistent with the disease. Subsequently, again in the month of May, further positivity was reported among the people living with the company workers. The mink farm in question has 19,500 adult animals (15,600 females and 3,900 males) and 73,200 young minks. It is an integrated cycle with a feed mill and annexed slaughterhouse, and it is the only American mink farm in the Autonomous Community of Aragon. With regard to the samples collected on the farm, four animals were sacrificed in May without finding a positivity to the virus (from lung, spleen, liver, and intestine). A weak positivity defined inconclusive by the Spanish health authorities was detected in June (1 out of 20 animals sampled with oropharyngeal swab). These samples were followed by further tests by the official control authority: serum samples and oropharyngeal and rectal swabs from 30 live animals and oropharyngeal and rectal swabs and lung parenchyma of six dead animals. One of the oropharyngeal swabs tested positive for SARS-CoV-2; in addition to this, seven other weak positivity were also observed in other animals, classified as inconclusive. In all cases the animals were asymptomatic. In July, however, the epidemic broke out on farm with 86.7% of swab-positive animals (90 rectal and oropharyngeal swabs in animals, of which 30 as adults and 60 as youngsters). Following this massive circulation of the virus, even in the absence of clinical signs compatible with the disease, the government of Aragon decided to kill all the animals on the farm with a destruction of the carcasses. No other cases were reported after this outbreak, but the dynamics of the circulation of SARS-CoV-2 in that company is significant, which seems to have taken 6 to 7 weeks from the first positive to obtain a circulation at the farm level.

Denmark: the first notification dates back to June 17 and the last update to November 5. Since June, in Denmark there has been a high circulation of the virus between minks with demonstrated transmission between minks and people. As of November 5, 207 different farms were involved. A policy of containment for the virus spread among mink farms has called for the killing and destruction of the skins of all the animals of the companies involved in positive cases and of neighboring companies within a radius of 7.8 km, which led to the destruction of 1.4 million minks. On November 4, the Danes announced the discovery of a mutation of the virus observed in mink, which was also found in the inhabitants who live near the farms. The Danish authorities noted that some of the mutations found in mink and in infected humans may reduce the effectiveness of antibodies naturally produced in previously infected people or some monoclonal antibodies under authorization [58]. For these reasons, the Danish authorities have decided to slaughter and destroy all minks in Denmark, including those in farms intended for animal reproduction. This resulted in the destruction of 12 million minks still in production despite previous containment measures introduced. This decision in fact destroyed the entire Danish industry of mink-skins production, because having destroyed all farms for reproduction, no farm will be able to restart for several years. In any case, it has been established that no farm will be able to reopen at least until 1/1/2022.

Italy: in the whole country there are only nine mink farms mainly distributed between four Regions: Lombardy, Emilia-Romagna, Veneto, and Abruzzo. These farms have been placed under health surveillance, with periodic visits by official veterinarians to the farms, since the Dutch authorities reported the first cases in their mink farms. On August 10, a virological positivity was observed in a farm (26,000 mink) located in the Cremona province, Lombardy: a human case was observed among the farm staff some days before. Restrictions on the movement of animals, sewage, and vehicles entering and leaving the farm were introduced. The positivity was observed following the transfer of 20 dead animals to the IZSLER in Brescia for the search of the virus. The analysis on this first group of animals tested revealed only one weak positivity, without pathological lesions attributable to respiratory disease. Following this positivity, the company was placed under strong clinical and virological surveillance, which involved a visit to the company every week, the collection of 30 throat and fecal swabs every two weeks, and the virological analysis of all dead animals for any cause. This involved the analysis of 1,124 between oropharyngeal and rectal swabs in live animals which showed only one other weak positivity, and no further positivity was observed from the 340 carcasses examined. On November 6, another rectal swab gave a weakly positive result in a different shed than the previous finding. Following this third weak positivity, the Ministry of Health, by order of November 22, 2020, decided to kill and destroy all animals on the farm and ordered the suspension of mink-farming activities on the national territory until February 28, 2021. Also in this Italian mink farm the SARS-CoV-2 had required months to be observed in different premises of the same farm.

Greece: Following the first confirmed case, which was reported on November 16, about 11 further cases of SARS-CoV-2 detection on mink farms were updated in a follow-up report on December 4. The first positivity was found following a report of clinical symptoms (inappetence, respiratory symptoms, and increased mortality) in a company of 2,500 adult minks located in the Kozani regional unit. Following the evidence of viral circulation, all the animals were slaughtered and the carcasses destroyed. A 10 Km sanitary cordon was established involving 40 other mink companies. With a view to One Health, Greece has decided to test all the care workers of the 92 mink companies in the area on a weekly basis, and any positivity in people has been communicated to the veterinary authorities. As a result of these investigation, another 11 positive companies were identified. In all cases, an epidemiological link with human cases was demonstrated. None of the amino acid mutations (Y453F, deletion of amino acids 69-70, I692V, M1229I), described on the rapid risk assessment of November 12, 2020, from EU agencies (ECDC, EFSA, EMA) regarding the Spike protein, was observed in the Greek minks and human SARS-CoV-2 sequenced genomes. Greek authority declared to not intend to cull other animals. Veterinary measures, restrictions, and strict biosecurity measures, including the mandatory use of PPE, have been imposed in all mink farms, which have been placed under official surveillance. Appropriately, the Greek Ministry of Health decided to include everyone who comes in close contact with minks (farm workers, owners, and veterinarians) in the highest priority group to receive vaccination.

Sweden: as of December 1, Sweden reported 13 mink farms with positivity in animals in Blekinge County in the southeast of the country. Twenty of the 40 Swedish mink companies are in this area. All farms detected as positive were observed in the aim of the surveillance, initiated in mid-October, and based on dead minks (aiming at 5 minks per week and farm) that must be submitted to SVA and encompasses all farms in the country. Dead minks from the farms were collected and sampled with swabs from the oral cavity and pharynx and analyzed with real-time RT-PCR. Notably, Sweden has decided not to stamp out animals at the infected farms at this point. This decision is supported by the small mink population, approximately 600,000 animals, of which around 80% already were planned to be killed for fur production during November and December [59]. The phylogenetic analysis of the strains found in Swedish minks and people in contact with groups in the same cluster confirmed their association (Lineage B1.1.39), and none of the mutations found in mink and Danish people have been observed in these viruses.

Lithuania: evidence of circulation of SARS-CoV-2 was established within the framework of passive surveillance due to increased mortality in a farm on November 26. An increased mortality rate was observed and involved 324 minks that were found dead. Randomly, 32 animals were sampled, and all were positive to SARS-CoV-2. In total, the virus was detected in five farm workers. Only the notification update is available for this country.

In conclusion of this part focused on the mink infection, we can summarize that as it happens for cats, dogs and ferrets, the virus arrived in mink farms brought by an infected and contagious person who looked after the animals. This epidemiological link was always observed everywhere. As can be seen in the cases described for example in Spain, Ital, and in The Netherlands, the introduction of the SARS-CoV-2 into the farm does not immediately lead to its spread among animals; however, this occurs after a certain time (up to 2 months), and an active circulation of the virus between animals was observed in different countries. Animals involved in circulation of the virus not only become infected, as often observed in dogs, but show clinical signs of the disease (inappetence, respiratory symptoms, and increase in mortality rate in the farm) and are able to transmit the virus to their similar, efficiently. Furthermore, in Denmark, this viral circulation has led to the selection of strains different from those that currently circulating most among the human population (D614G). In fact, in Denmark at least five viral clusters associated with mink have been observed, which infected 214 people among farm staff and people who lived near the farms but did attend the farms. One of these groups, called cluster 5 which circulated in Denmark between August and September, has four mutations (three substitutions Y453F, I692V, and M1229I and a deletion del 69 / 70) in the virus spike protein. This particular conformation of the spike was observed in 12 out of 214 people infected with variants adapted to mink. Of these mutations, two (Y435F and N439K) fall into the RBD region of the spike protein (RBD is a region required for the first binding of SARS-COV-2 protein S to the human ACE2 receptor). Starr and colleagues discovered that Y435F increased the spike-binding capacity to human ACE2 [60]. However, these experiments were performed using the old spike protein of an early SARS-CoV-2 strain with the motif D614. According to the UK new and emerging respiratory virus threats advisory group (NERVTAG) [61], the D614G mutation dominant in human strains already possesses increased affinity for human ACE2, and it is possible that the addition of Y453F now makes little difference to the interaction. Nevertheless, outbreaks in Danish mink farms have the D614G mutation and were still selected for Y435F during farm circulation in minks. So that these mutations appear to be additive and successive. It is, in fact, epidemiologically reasonable to assume that the strain that entered the mink farms carried by humans had the D614G mutation and the Y435F mutation emerged and was subsequently selected in minks.

Moreover, it should be noted that the viral variant called Y435F, defined as the variant associated with mink, has already been observed outside of Europe (South Africa in June 2020, Russia and Australia in August 2020, and the USA in October 2020), so the containment of this viral variant may already be seriously compromised. In any case, the ECDC (European Center for Disease Prevention and Control) in the document RAPID RISK ASSESSMENT "Detection of new SARS-CoV-2 variants related to mink" of November 12, 2020, has assessed that the risk of infection from mink-associated strains is low for the general population, moderate for populations in areas with a high concentration of mink farms, and very high for people working on mink farms (including veterinarians).

## 5. Discussion

SARS-CoV-2 can infect and sometimes cause diseases in several animal species. Among all, those closest phylogenetically to humans (gibbon, yellow-green vervet, macaque, orangutan, and chimpanzee) and possessing an identical virus receptor to ours are certainly the most susceptible to infection and disease. Among other species, felines have certainly shown both in laboratory studies and in real cases reported to be susceptible to the virus and sometimes able to transmit it between animals. Moreover, frequently the virus among felines caused observable symptoms, that are mild in most cases (sneezing, loss of appetite, apathy, weakness, and sometimes ocular discharged was observed), while in few cases, severe respiratory symptoms with bilateral pneumonia and breathing difficulties were reported. Dogs are the only species of Canidae that was reported as infected by SARS-CoV-2, and in the large majority of the cases, no signs of disease were observed in dogs affected. Only in the USA, passive surveillance identified not specific symptoms in older animals and with various comorbidities not permitting to understand the role of the virus in dogs.

It is absolutely important here to reiterate that in all documented cases in which the virus has infected domestic animals, this happened through the human–animal route. Therefore, pets do not represent a risk for the transmission or maintenance of the virus in the human population at the present state of knowledge, and it is mandatory to avoid what has happened in China, where a severe public response that resulted in many pets, dogs, and cats, being killed or abandoned, following the public statement of a member of the senior expert team from China’s National Health Commission that pet owners should take extra care of their animals [62].

Mustelids deserve a separate and different discussion. As observed in experimental infection studies, ferrets once infected with the virus develop signs of respiratory disease and effectively transmit the virus to other ferrets [41]. Minks farms in the cases reported have been infected with the virus almost everywhere in Europe and the USA. It is interesting to note that epidemiological descriptions of the outbreaks often allow us to hypothesize a certain latency period between the entry of the virus on the farm and its active circulation among the animals. In the Dutch cases reported, where strict veterinary surveillance was implemented, in 60% of cases, the positivity was observed in animals that died from all causes analyzed as part of the surveillance, without any signs of the disease being observed on the farm. While in the remaining 40%, the cases were identified after the observation of signs of disease on the farm (increased mortality, lack of appetite, etc.). In addition, the serological investigation in animals sometimes observed up to 70% of animals with detectable antibodies, testifying to a massive, and not recent, circulation among the animals of the farm once identified. In the case described in Spain, the first weak positivity that were defined as inconclusive were observed in the second half of May. The virus circulated under trace in the farm for at least 6 weeks, until July when the epidemic broke out on farms with 86.7% of swab-positive animals. A similar dynamic was observed in Italy, where less diffusion of the virus was observed in the involved farm with only few samples that tested positive. Also in this low-circulation scenario, positivity in different premises were observed and in those cases required almost three months. Coronavirus circulation among animals inevitably leads to observed mutations in the virus. The spike protein of the virus is the most affected by these mutations and is also one of the most subjected to natural selection. The circulation of SARS-CoV-2 in mink farms in Denmark led to the emergence of a new variant of the virus called Y435F, which was defined as the variant associated with mink, that has been observed also in human populations that live around the mink-farm district in Denmark. However, the Y435F mutation was also already observed in strains unrelated to the Danish variants in other countries (the Russian Federation, South Africa, Switzerland, and the United States) in the GISAID EpiCoV sequence database outside of Europe [63]. Randomly acquired genetic changes when the virus replicates are common in coronaviruses, due to their large genome, the infidelity of RdRp enzyme, and selection pressure during the adaptation of the virus to new conditions, such as a new host [64]. Furthermore, homologous recombination within viral structural proteins between coronaviruses from different hosts may be responsible for cross-species transmission. These features could allow the virus to become endemic in some farmed and wild animal populations. In agreement with [9], the SARS-CoV-2 pandemic and the consequent panzootic potential highlight the need for a One Health approach. Once the reverse spillover events occur, it is crucial to know whether infection is maintained in an animal population and has a potential to spillover back to humans. As recently reported by [65], a One Health approach requires the creation of a network capable of merging the contributions of different areas of expertise. People working in contact with animals should be provided with all appropriate safety equipment thus reducing the human-to-animal viral diffusion. Furthermore, testing of workers, contact tracing, isolation, and quarantine should to be immediately initiated when a human case is related to an animal farm, and this approach should not be limited to highly susceptible farmed animals such as mink. Hunting and wildlife rearing put people in contact with wild animals, and often those animals suffer from debilitating and immunocompromising conditions that promote zoonotic outbreaks [65]. Veterinary surveillance of zoonotic diseases must therefore be maintained and implemented where not present with the aim of early detecting the presence of SARS-CoV-2 in animals. The isolated strains of SARS-CoV-2 must be sequenced systematically, and the sequences obtained from infected animals should be made available to the scientific community, in order to monitor the emergence or spread of variants of the virus adapted to animals but still potentially dangerous for humans. Especially now, with vaccination becoming available in the human population, veterinary surveillance among susceptible species and other animal species, including wild animals, with particular attention to wild mustelids, will have to be strengthened to avoid considering prematurely to have defeated a virus that has only hidden itself among animal species.

## Figures and Tables

**Figure 1 life-11-00123-f001:**
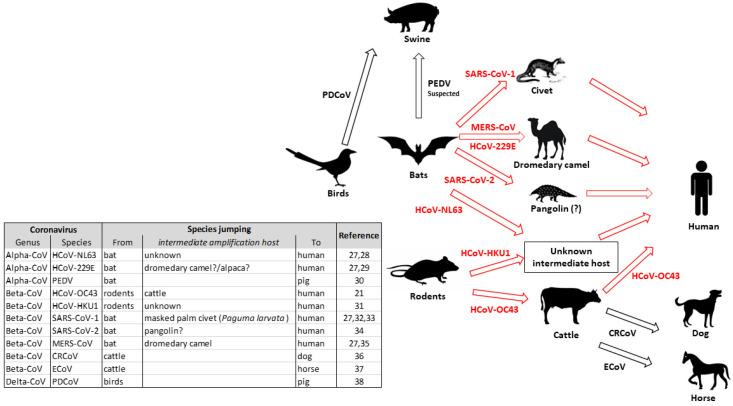
Representation of interspecies jumping of Coronaviruses. Red arrows indicate cross-species transmission involving an intermediate (proved, suspected or undetermined) amplifying host. Black arrows represent a direct transmission between two species [21,27,28,29,30,31,32,33,34,35,36,37,38].

**Table 1 life-11-00123-t001:** List of coronaviruses affecting farmed and companion animals, including proved natural infection by SARS-CoV-2.

Animal Species	Coronaviruses	SARS-CoV-2
American Mink (*Neovison vison*)	MCV						Yes
Cat (*Felis catus*)	FCoV-I	FCoV-II					Yes
Cattle (*Bos taurus*)	BCoV						
Chicken (*Gallus gallus domesticus*) and other birds	IBV	IBV-like CoVs					
Dog (*Canis lupus familiaris*)	CCoV-I	CCoV-II	CRCoV				Yes
Ferret (*Mustela putorius furo*)	FRCoV						
Goat (*Capra hircus*)	BCoV-like CoVs						
Guineafowl (fam. *Numididae*)	GfCoV						
Horse (*Equus ferus caballus*)	ECoV						
Pheasant (*Phasianus colchicus*)	PhCoV						
Swine (*Sus scrofa*)	TGEV	PEDV	PRCoV	PHEV	PDCoV	SADS-CoV	
Rabbit (*Oryctolagus cuniculus*)	RbCoV						
Sheep (*Ovis aries*)	BCoV-like CoVs						
Turkey (genus *Meleagris*)	TCoV						
Water Buffalo (*Bubalus bubalis*)	BuCoV

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
