# Peer review of "Animal Coronaviruses and SARS-COV-2 in Animals, What Do We Actually Know?"

_life, 2021, doi:10.3390/life11020123_

Round 1

Reviewer 1 Report

In this manuscript,  the authors summarize what is known at present on the Coronaviruses of interest for veterinary medicine on the basis of their host animal species.  In particular, they focus on SARS Cov-2 infection in farm and domestic animals by analyzing both research results and official documents notified to OIE by different countries where the animal infections have been registered. All this information further confirms the human-animal route, and that in the majority of the cases the animals are asymptomatic or show mild symptoms. The only case of transmission from an animal species Covid-19 positive to humans regards the domestic mink, registered in those countries that housed the farms.

What the authors observe is that the serological tests show that the animals may have an incubation time longer than that observed in humans. Thus, the serological tests can help in determining the presence of the virus in a farm even in the absence or late onset of symptoms in the animals,  stressing the need for strict veterinary surveillance, according to the criteria of One Health, of the domestic, farm, wild animals and in particular of the mustelid species in which the SARS-CoV-2 changes and the minks become able to transmit it to humans again. 

I think the manuscript is quite interesting, however, I think the One Health concept should be a little in-depth: please see:  Costagliola A, Liguori G, d'Angelo D, Costa C, Ciani F, Giordano A. Do Animals Play a Role in the Transmission of Severe Acute Respiratory Syndrome Coronavirus-2 (SARS-CoV-2)? A Commentary. Animals (Basel). 2020 Dec 24;11(1): E16. DOI: 10.3390/ani11010016. PMID: 33374168.

I also encountered a couple of typos that need to be corrected. 

line 126: belongs

line 402: reports

Author Response

The Authors would like to thank the reviewer for reading the manuscript and kind judgment on whether it may be of interest. The authors care a lot about the concept of One Health and therefore have welcomed the suggestion to deepen the concept and thank for the suggestion. Please review the discussion section for the point covered (Lines 599-613 in the revised manuscript).

line 126: belongs - done thanks

line 402: reports - done thanks 

Reviewer 2 Report

The review give an overview about animal coronaviruses and infection of animals with the SARS-CoV-2. There are many review articles about such subject. I have the following major suggestions:

  1. A table containing the list of Coronaviruses in animals will increase the feasibility of the review article. Another table with the list of reports of animals infected with SARS-CoV-2 is necessary. 
  2. A graph of cross-species jump of coronavirus must be created. Like BCoV to CRCoV and FCoV to CCoV. 
  3. A world map with reported animal cases is missing in the article. 
  4. A table of the presence of ACE-1 receptors in animal must be included. 

Author Response

The authors thank the reviewer for helpful suggestions. 

  1. A table containing the list of Coronaviruses in farmed and companion animals was added (Table 1 in the revised manuscript). 
  2. A table with the list of reported cases and also the world map with reported animal cases are available on OIE web site, so we decided not to insert a table in the text that could not be updated after publication. However, we now referred in the text to the existence of this schematic representation, thanks (see Lines 252-256 in the revised manuscript). 
  3. A graph of cross-species jump of coronavirus was added in the text (Figure 1 in the revised manuscript). Authors thank the reviewer for this suggestion because the graph highlights the plasticity of coronavirus genome and their cross-species transmission potential
  4. A table of the presence of ACE receptors in animals was de facto difficult to be inserted in the text because all mammalian species have the receptors for the virus (ACE2) with important differences between species that are summarized in the text.  

Reviewer 3 Report

The proposed review does not bring real novelty. The form is mainly that of a catalog and an administrative report. There is no thorough analysis. The topic is very interesting but the way it is addressed is rather disappointing. What one would have expected is an analysis of how SARS-CoV-2, and other viruses, can disseminate in the wild, circulate from one species to another including humans. One also expects hypotheses on the routes of dissemination and infection, the biological mechanisms involved and an assessment of the risks.

Unfortunately, under this form the manuscript cannot be accepted for publication. 

Author Response

Dear Value Reviewer, we respect your comments but disagree with your description of our proposal. Our manuscript represents a review work and has thus been proposed to the editors of the journal. A review must not contain elements of novelty, but a rigorous and complete analysis of the knowledge available on a specific topic.   Honestly, what you asks to discuss is already explained in the text at various points, and please consider that some parts have been further implemented in the revision that we have already prepared after receiving the two previous comments.   In particular, the analysis of the species theoretically susceptible to the virus from the point of view of biological mechanisms (the presence and conformation of the ACE2 receptor) is present. An in-depth analysis of the real cases and of the serological investigations carried out was certainly present in the manuscript and some new aspects were deduced (as observed by reviewer 1). Finally, the analysis of the risks associated with the spread of the virus among animals and the potential pan-zootic risk was present and was further implemented after the review.   Impossibly to answer in point by point to your review, please consider our revised manuscript where we have added table and figure that could clearly explain the relationship between different coronaviruses and their ability to spread between different species.    

Round 2

Reviewer 2 Report

The authors has improved the manuscript and included all necessary changes. The review will be of added value to understand CoV in animals. 

Author Response

the authors would like to thank the reviewer for contributing to the improvement of our manuscript.